# Farmers' Perception of Climate Change and Climate-Smart Agriculture in Northern Benin, West Africa

**Fidèle T. Moutouama** [1,*] **, Ghislain T. Tepa-Yotto** [1,2] **, Cyriaque Agboton** [1] **, Brice Gbaguidi** [1] **, Haruna Sekabira** [3] **and Manuele Tamò** [1]

[1] Biorisk Management Facility (BIMAF), International Institute of Tropical Agriculture (IITA-Benin), 08-BP 0932 Tri Postal, Cotonou 01000, Benin; g.tepa-yotto@cgiar.org (G.T.T.-Y.); c.agboton@cgiar.org (C.A.); b.gbaguidi@cgiar.org (B.G.); m.tamo@cgiar.org (M.T.)

[2] Ecole de Gestion et de Production Végétale et Semencière (EGPVS), Université Nationale d'Agriculture (UNA), Ketou 43, Benin

[3] International Institute of Tropical Agriculture (IITA-Uganda), Kampala 28565, Uganda; h.sekabira@cgiar.org

* Correspondence: fidelemoutouama@gmail.com

**Abstract:** Climate-Smart Agriculture (CSA) is an approach that identifies actions needed to transform and reorganize agricultural systems to effectively support agricultural development and ensure food security in the face of climate change. In this study, we assessed farmers' perception of climate change, available CSA practices (CSAP) and the determinants of CSAP adoption in northern Benin. A list of CSAP was generated from a workshop with different stakeholders. Face-to-face interviews were then carried out with 368 farmers selected based on stratified random sampling in the study area. Binomial generalized mixed-effect models were run to analyze the relation between socio-demographic characteristics and the use of CSAP. CSAP were evaluated using a three-point Likert scale and the frequency of agreement with the statement that the selected practices meet the pillars of CSA. More than 60% of farmers had heard about climate change, and more than 80% had observed changes in temperature, rainfall amounts and distribution. Thirty-one CSAP were identified in the area, and only 11 were known by more than 50% of farmers. Out of the 12 selected CSAP for the assessment of adoption and evaluation, seven (7) were used by more than 50% of those who knew them. Farmers agreed with the statements that the evaluated practices improved farm productivity and adaptation to climate change but did not mitigate climate change. Ethnic group and education level were the two major factors that significantly determined the use of the evaluated CSAP.

**Keywords:** climate-smart; Benin agroecological zone IV; adoption; agriculture

## 1. Introduction

Climate variables relevant to food security and food systems are predominantly temperature and precipitation-related, but they also include integrated metrics that combine these and other variables such as solar radiation, wind, and humidity [1]. According to the IPCC report [1], the impact of climate change through changes in these variables is projected to negatively impact all aspects of food security (food availability, access, utilization, and stability). Without appropriate interventions, climate change and variability will affect agricultural yields, food security and add to the present unacceptable levels of poverty in sub-Saharan Africa [2].

With a population increase rate of 2.7%/year [3] and 54.8% of its work force being in the agricultural sector [4], Benin is the 13th most vulnerable country and the 55th least ready country with regard to climate change [5]. It has both a great need for investment and innovations to improve readiness and a great urgency for action. Investments and innovation in the sector of agriculture will contribute to reverse the country's high vulnerability and increase communities' resilience to climate variability [6].

According to Benin's third communication to United Nation Framework Convention on Climate Change (UNFCC) [7], Benin's future vulnerability assessment (by 2050) foretells incertitude regarding variables (rain and temperature) that are essential for crop production in a traditional rainfed agriculture system. The changes in rain patterns, temperatures and ecosystem features will further reduce land productivity and jeopardize communities' well-being. This is more obvious for the poorest due to their heavy reliance on nature and low resilience to climate disturbance.

According to Benin's third communication to UNFCC, in 2015, agriculture was second after energy in terms of Greenhouse Gas (GHG) emissions with 41% of the total direct emission (Forestry excluded) [7]. Consequently, any solutions to reduce climate change impact on agriculture should go beyond improving and securing productivity by minimizing the effect of agriculture on climate as well.

Climate-Smart Agriculture (CSA) is an agriculture approach that sustainably increases productivity, improves resilience (adaptation), reduces/removes GHGs (mitigation), and enhances the achievement of national food security and the sustainable development goals [8]. Across Africa, farmers are embracing "climate-smart" innovations that could help fuel a dramatic increase in food production despite an increasingly challenging agriculture environment [9]. In addition, many studies have been carried out to explore issues around climate change and examine various perspectives and lessons learned on technologies and practices through a CSA lens [10]. Adaptation measures that have been successfully tested for wide application within a given region should be scaled up, depending on the context of the country, while taking agro- ecological zones into account [11]. Indeed, CSA is a location-specific, knowledge-intensive approach, so it becomes necessary to identify adoption barriers to better target appropriate solutions [12].

The current study aims at (i) assessing farmers' perception of climate change and available Climate-Smart Agriculture Practices (CSAP) and (ii) assessing the relationships between some socio-demographic factors and the use of CSAP.

The presumed relationships between climate change, CSA and farmer's perception and adoption of CSAP are shown in Figure 1.

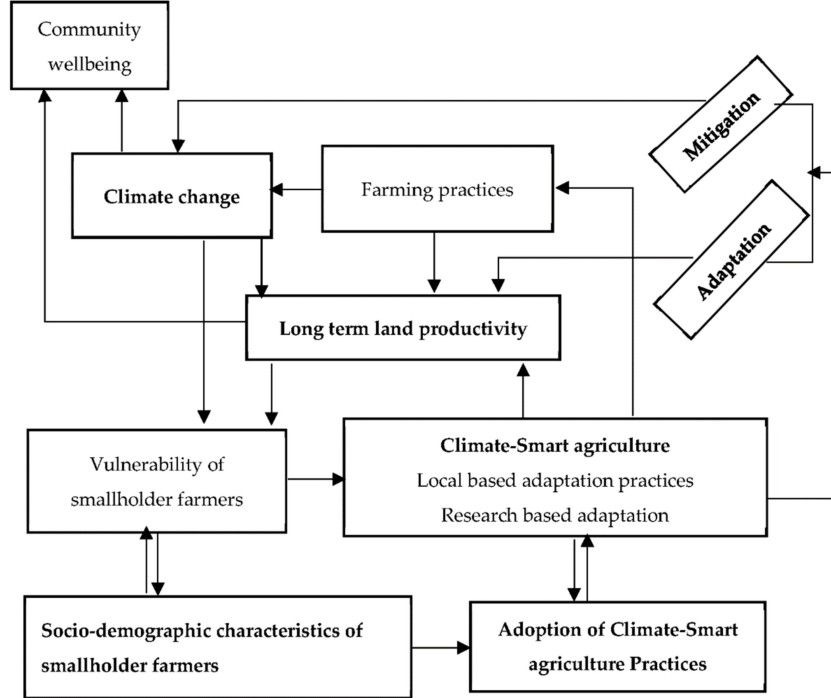

**Figure 1.** Conceptual framework showing presumed relationships between climate change, CSA and farmer's perception and adoption of CSAP. Sources: authors.

Climate change and inappropriate farming practices affect long-term land productivity. Climate change and land productivity affect smallholder vulnerability, which is also affected by farmers' socio-demographic characteristics. In response to farmers' vulnerability, Climate-Smart Agriculture (CSA) has been developed. CSA adjusts farming practices to improve land productivity while mitigating Greenhouse Gases emission. Climate change impacts are then reduced through the mitigation and improvement of land productivity. The rate of adoption can be influenced by the socio-demographic characteristics of smallholder farmers and the nature of the practice. When a practice is adopted and is acknowledged to be effective, other farmers follow the example.

## 2. Materials and Methods

### 2.1. Study Area

The study was carried out in northern Benin, specifically in Agroecological Zone IV (AEZ IV), which is also known as the West Atacora Zone (Figure 2). It stretches from North Donga to West Atacora and encompasses nine (9) municipalities of which eight (8) were considered in this study (Tanguiéta, Cobly, Matéri, Toucoutouna, Natitingou, Boukoumbé, Copargo and Djougou). The zone lies between 9°16′00″ and 11°27′20″ North latitude and 0°45′00″ et 2°12′10″ East longitude and falls in the Sudan transition zone.

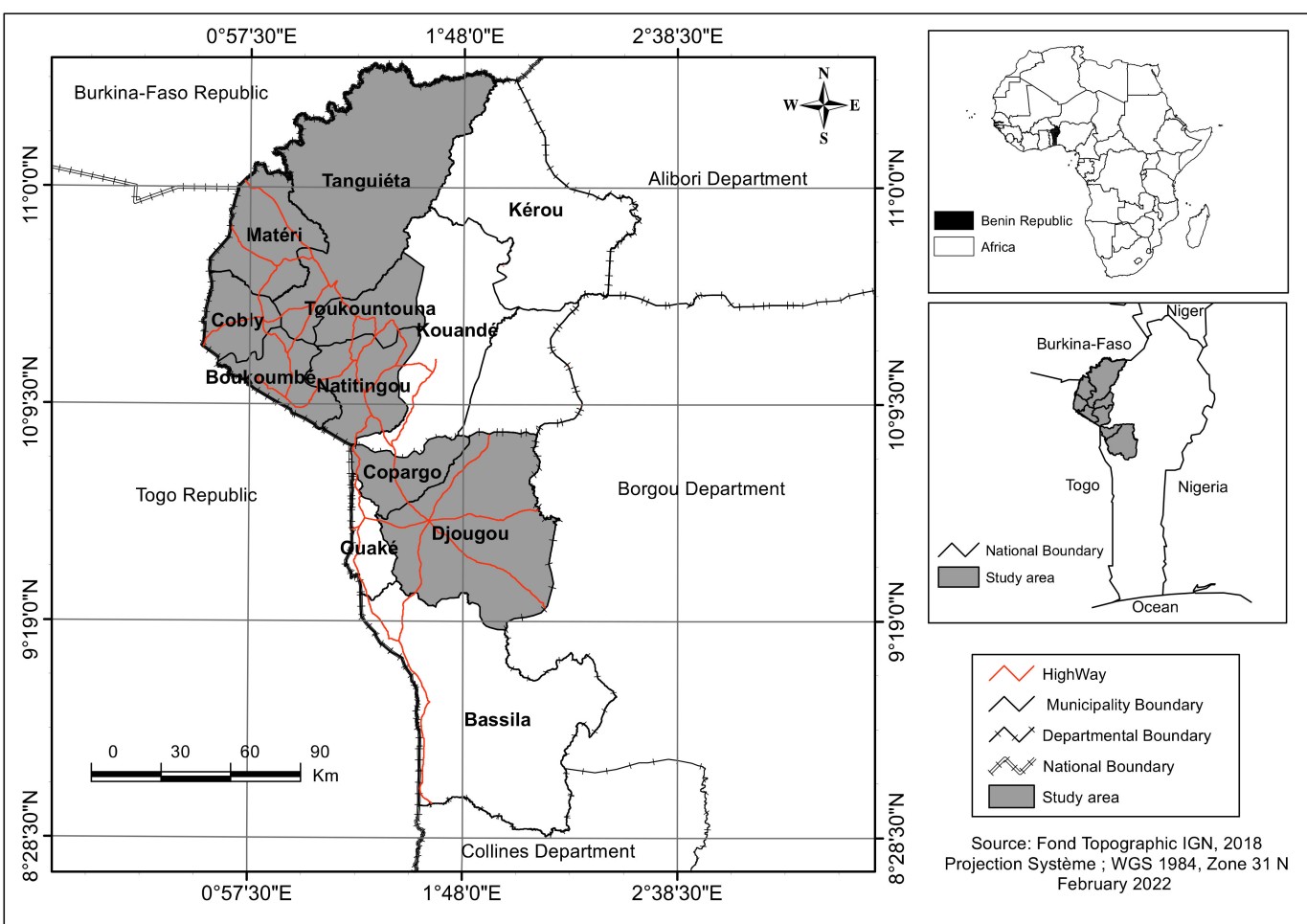

**Figure 2.** Study area.

The AEZ IV was purposely selected due to its unique landscape and the potential climate change impacts. Indeed, the area is characterized by the presence of the Atacora Chain of Mountains, three rivers (Ouémé, Pendjari and Mékrou) and the Pendjari National Parc. The climate is tropical of a Soudanian type with one rainy season (April–October)

and one dry season (November to April). The Atacora Chain of Mountains creates higher rainfall in some municipalities as compared to other similar climatic zones, with annual rainfall ranging from 1200 to 1350 mm. Another specificity of the area is the Harmattan, which is a wind blowing from the Sahara in a northeast direction.

The AEZ IV faces some natural constraints, including land and water availability. The presence of mountains, the National Pendjari Parc and the stony soils reduce the available farmland. Moreover, the inappropriate cropping systems have exhausted soils, leading to a continuous soil fertility decrease. In addition, the uneven topography and the high-intensity rains observed in the area erode the land and cause serious soil degradation issues. The long dry season prevents smallholder farmers, who depend on rainfall, from growing crops year-round.

Benin's third national communication to UNFCC provides how Benin's future climate will be and the consequences on crop production. Projections indicated that by 2050, precipitations will vary either positively or slightly negatively depending on the models. All scenarios agree that Benin will become warmer (0.8–2.3 °C). The uncertainty of future rainfall and the change in temperature would cause a reduction in crop productivity (30% for maize and 20% for cotton) [13]. Two of the municipalities (Copargo, and Djougou) involved in this study have been predicted by simulations to be the most vulnerable by 2050 in terms of the number of years with decrease in rainfall [13].

*2.2. Data Collection*

Farmers' perceptions and evaluation of CSAP were assessed through face-to-face interviews with 368 smallholder farmers in eight (8) municipalities using the stratified random sampling method. The questionnaire was administered to the heads of households.

Prior to this, a workshop was held with the various stakeholders (25 people from agriculture sector, farmers associations, NGOs and research institutions) to obtain a list of CSAP for the zone. At the workshop, the concept of CSA was presented, and the participants were put in groups (3 balanced groups) to list the CSAP observed in the areas along with the climate risk these practices contribute to tackle. The lists were all presented at the plenary for validation and combined with those CSAP and technologies established by FAO [14] for Benin which were effectively in use in the area. The practices that the participants did not mention but agreed upon existing in the area were also added to make the final list for the whole area. This list was used to assess farmers' awareness of climate change, their perception of the identified CSAP, their evaluation of the practices and what drives the non-use of these practices.

CSAP were evaluated by smallholder farmers based on the three pillars of Climate-Smart Agriculture: productivity, adaptation, and mitigation, which were explained to them as summarized below:

1.　Productivity: CSA sustainably increases productivity and incomes and positively affects food security.
2.　Adaptation/resilience: CSA reduces vulnerability to drought, pests, disease and other climate-related risks and shocks. It improves the ability to adapt and grow in the face of long-term stresses such as shortened seasons and irregular weather conditions.
3.　Mitigation: CSA strives for lower emissions for every calorie or kilogram of food produced, avoids deforestation from agriculture, and identifies ways to absorb carbon from the atmosphere.

Wijk et al. [15] provide some elements to improve the assessment of the three pillars of CSA. The authors indicate that sustainability is key to productivity and food security assessment and adaptation should be assessed for the long term. In this study, all the practices and technologies collected at the workshop were listed, and the farmers were asked whether they know or do not know the practices. The evaluation of the practices was performed for the most known practices among farmers, having scored more than 50%. These practices were then evaluated through their performance with the three pillars. For each practice, a statement was made that the practice meets a pillar of CSA for the long

term. For example, a farmer using "practice n" will be asked to choose in the three-point Likert scale [16] after he had listened to the following statement: "'Practice n' has increased the productivity of your crop since you have been using it and contributed to increase your income". Productivity, adaptation, and mitigation were explained to farmers in the local language using practical examples of proxies measuring each of the pillars as included in the above definition of each pillar. The three-point Likert scale was appropriate to make it simple for farmers who have a simple understanding of CSA pillars and do not have a measured appreciation of its pillars.

### 2.3. Data Analysis

The choice of adopting a technology is preceded by a number of mental processes which include: awareness, interest, evaluation, trial, and adoption [17]. According to the encyclopedia of qualitative research methods [18], perception is like a set of lenses through which an individual views reality. In this study, the perception of climate change was assessed through the frequency of "awareness" or "knowledge" of climate change and how the interviewees observe the changes in temperature, rainfall, and spatial–temporal distribution of rain. The practices involved in the study were not necessarily promoted by an organization. Then, "the adoption of a practice" is hereby assessed through "the use of the practice".

To test the effect of socio-demographic variables (age, ethnic group, education level, household size) on local farmers' perception of climate change impact on agriculture, we performed binomial generalized mixed effect models using the package glmmTMB [19]. We chose these models for two reasons. First, our response variables here (temperature change perception, rainfall change perception, awareness of climate change, use of practice) are all binary variables (i.e., yes, or no). Second, our data have a hierarchical structure. Here, each department was considered as a random variable and the other variable was considered as fixed. To evaluate each of the practices known by more than 50% of the interviewees, the frequency of agreement that the practice meets the definition of the pillars was considered. All the analyses were performed in R 3.6.2 [20].

## 3. Results

### 3.1. Climate Change Perception

The perception of climate change was assessed through the main climate parameters that farmers can easily understand. More than ninety-eight percent (98.4%) of the surveyed farmers recognized that the temperature in the study area is changing, and 92.9% observed that the weather is becoming warmer. Almost all the surveyed farmers (99.7%) acknowledged that rainwater quantity has changed compared to the previous years and for 98.1% of them, the weather is becoming drier. In addition, 88% of the surveyed farmers think rainfall comes later than in previous years. Nevertheless, a high percentage (88%) of farmers are not able to forecast how the coming season is going to be. Sixty-nine percent (69.7%) of the farmers have heard about climate change phenomenon through media, NGOs, other farmers, their neighborhood, meetings, agroecology trainings, local agricultural services, and traditions. The major means of information are media (33.7%), meetings (10%), NGOs (9.2%), tradition (5%) and the neighborhood (4.2%). The results of binomial generalized mixed effect models showed that no socio-demographic factor shapes farmers' perception of climate change in Benin AEZ IV.

### 3.2. Awareness of Climate-Smart Agriculture Practices and Determinants of Adoption

Table 1 shows the frequencies of awareness and use of CSAP. Thirty-one (31) Climate-Smart Agriculture Practices (CSAP)/technologies were compiled from the workshop (see Table A1 for description).

**Table 1.** Frequencies of awareness and use of CSAP.

| Practice Number | Practice Name | Percentage of Knowledge | Percentage of Use |
|---|---|---|---|
| P1 | Spiritual invocation of rain | 59.5 | 23.2 |
| P2 | Climate risk forecast/early warning system | 25.1 | 9.8 |
| P3 | Crop rotation | 81.8 | 89.9 |
| P4 | Sowing spread over time | 31.3 | 51.7 |
| P5 | Reduction in seed density | 50.0 | 46.1 |
| P6 | Use of improved varieties | 77.1 | 73.5 |
| P7 | High number of seeds per pocket followed by thinning | 48.9 | 51.7 |
| P8 | Irrigation | 25.7 | 2.8 |
| P9 | Cropping on lowlands (marsh), flooded areas and riverbanks | 65.1 | 70.1 |
| P10 | Half moons | 0.6 | 1.4 |
| P11 | Ploughing perpendicularly to the slope | 1.1 | 33.0 |
| P12 | Fence made of tree branches | 0.8 | 7.0 |
| P13 | Use of Zai | 28.2 | 14.3 |
| P14 | Staggered plowing | 41.6 | 31.0 |
| P15 | Stone rows | 33.5 | 19.3 |
| P16 | Double plowing | 52.5 | 41.9 |
| P17 | No-tillage | 60.9 | 55.3 |
| P18 | Mulching | 21.8 | 17.9 |
| P19 | Sowing under plant cover | 17.3 | 10.9 |
| P20 | Association of crops with Pigeon pea (*Cajanus cajan*) | 27.9 | 18.4 |
| P21 | Association with mucuna or *Aeschynomene histrix, Stylosanthes guianensis* | 19.6 | 2.8 |
| P22 | Association Yam and *Gliricidia* | 14.5 | 2.0 |
| P23 | Cultivation of less water-intensive vegetables | 22.9 | 13.7 |
| P24 | Use of agricultural residues | 33.5 | 5.6 |
| P25 | Use of funnel, animal droppings | 58.4 | 24.9 |
| P26 | Use of compost | 40.8 | 14.5 |
| P27 | Direct parking of oxen in the fields | 65.6 | 20.1 |
| P28 | Use of mineral fertilizers | 78.5 | 69.8 |
| P29 | Use of micro-dose | 41.1 | 34.4 |
| P30 | Use of biopesticides | 45.3 | 38.3 |
| P31 | Using ash against attacks | 58.7 | 22.9 |

Crop rotation appeared to be the most known (81.8%) and at the same time the most used (89.9%) practice in the study area. Apart from crop rotation, mineral fertilizer, improved varieties, direct parking of oxen in the fields before ploughing, exploitation of lowlands and flooded areas and no-tillage were known by more than 60% of the interviewees (78.5%, 77.1%, 65.6%, 65.1% and 60.9%, respectively).

Eleven (11) practices were known by more than 50% of the interviewees. Only three practices were known by less than 14%. These include fascines, half-moon and ploughing perpendicularly to the slope with 0.8%, 0.6% and 1.1% of awareness rate, respectively. The

first three practices, crop rotation, mineral fertilizers and improved crop varieties, also showed the highest rates of use (89.9%, 69.8% and 73.5%, respectively).

No CSAP was used by all the people that knew them. The rate of use ranges from 1.4% (for half-moon) to 89.9% (for crop rotation). Out of the 31 practices, only seven (7) were used by more than 50% of the people knowing them: sowing spread over time (51.7%), high number of seeds per pocket followed by thinning (51.7%), no-tillage (55.3%), mineral fertilizers (69.8%), cropping on lowlands (marsh), flooded areas and riverbanks (70.1%), improved varieties (73.5%), and crop rotation (89.9%).

Some well-known practices have relatively low rates of use. Such practices include the direct parking of oxen, spiritual invocation of rain, and the use of ash to control pests.

Results of the binomial generalized mixed effect models (Tables A2 and A3) revealed that the use of CSAP in the study area is affected by farmers' ethnic group and their education level. Out of the 12 practices which were subjected to evaluation and determinants assessment, farmers' ethnic groups determined the use of eight practices: spiritual invocation of rain, reduction in seed density, high number of seeds per pocket followed by thinning, cropping on lowlands (marsh), flooded areas and riverbanks, no-tillage, direct parking of oxen in the fields, mineral fertilizers and using ash against attacks. Farmers' level of education affects the use of spiritual invocation of rai, reduction in seed density and using ash against pests.

### 3.3. Evaluation of Climate-Smart Agriculture Practices

The evaluation of CSAP was carried out with the twelve first practices which were known by at least 50% of the interviewees except for the use of a high number of seeds per pocket followed by thinning with less than 50% (Table 1). This practice was included because its rate was very close to 50 (48.9%).

Following farmer's evaluation (Table 2), crop rotation, direct parking, animal droppings, improved variety and mineral fertilizer were the first five practices that improve crop productivity. For adaptation, animal droppings, improved varieties, double plowing, cropping on lowlands, flooded areas, and riverbanks and spiritual invocation of rainfall appear to be the first five practices. Only animal dropping has a high frequency of agreement (95.4%) with the statement that it contributes to mitigation. Crop rotation appears second with 48% frequency of agreement. For all other practices, the frequency of disagreement with the statement that the practices improve mitigation was higher than that of agreement. After the use of animal droppings and crop rotation, no-tillage, direct parking of oxen in the fields and the use of improved varieties were the following practices with higher frequency (41.7%, 33% and 32.8%, respectively).

**Table 2.** Classification of the top 5 CSAP by the frequency of agreement for each CSA pillar.

| Rank | Productivity | Adaptation | Mitigation | Most Known | Most Used |
|---|---|---|---|---|---|
| 1 | Crop rotation | Use of funnel, animal droppings | Use of funnel, animal droppings | Crop rotation | Crop rotation |
| 2 | Direct parking of oxen in the fields | Use of improved varieties | Crop rotation | Use of mineral fertilizers | Use of improved varieties |
| 3 | Use of animal droppings | Double plowing | No-tillage | Use of improved varieties | Cropping on lowlands (marsh), flooded areas and riverbanks |
| 4 | Use of improved varieties | Cropping on lowlands (marsh), flooded areas and riverbanks | Direct parking of oxen in the fields | Direct parking of oxen in the fields | Use of mineral fertilizers |
| 5 | Use of mineral fertilizers | Spiritual invocation of rain | Use of improved varieties | Cropping on lowlands (marsh), flooded areas and riverbanks | No-tillage |

Figure 3a shows the frequency of farmers agreeing with the statement that each CSAP increases crop productivity and income for more than one season. More than 80% of farmers agreed that almost all the practices contribute to improve their crop productivity and income. The lowest frequency was observed with the practice no-tillage (P17 with 58%). Almost all farmers (more than 99%) agreed that crop rotation (P3) and direct parking of oxen (P27) improve crop productivity and household income. More than 90% of the interviewees also agreed that improved varieties (P6), cropping on lowlands (marsh), flooded areas and riverbanks (P9), double plowing (P16), animal droppings (P25) and mineral fertilizers (P28) positively affect productivity and income.

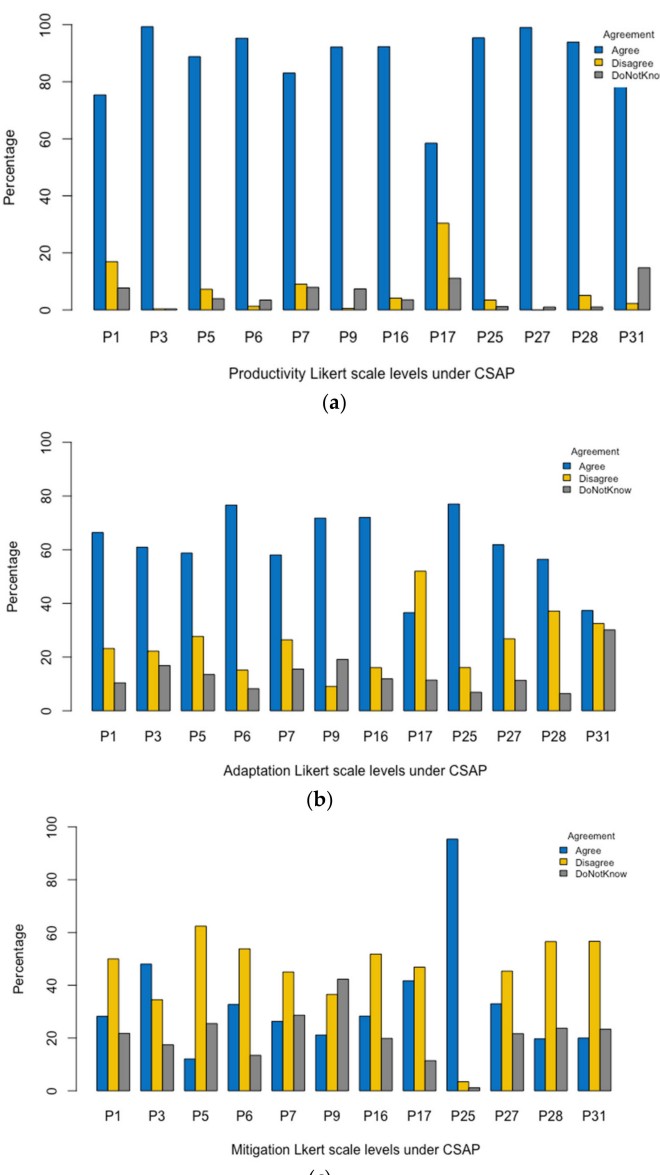

**Figure 3.** Frequency of Likert scale levels of agreement with the statement that Climate-Smart Agriculture Practices (CSAPs) contribute to CSA's pillars: (**a**) Productivity and income; (**b**) Adaptation; (**c**) Mitigation.

The percentages of farmers who disagreed or do not know the effectiveness of the 12 practices with regard to productivity and income increase were low.

Figure 3b shows the frequency of farmers agreeing with the statement that each CSAP improves crop resistance to drought, pest attacks and disturbance in rainfall distribution (adaptation). The frequency of agreement was higher than that of disagreement for all the

practices except for no-tillage (P17) for which 52% of people disagree with the statement that it improves crop resistance to drought and disturbances in rain distribution. Using ash against insects' attacks (P31) also had a low level of agreement. Among farmers who use the practices, up to 32.5% disagree that the practice improves adaptation and 30.1% do not know the practice's effectiveness.

Figure 3c shows the views of farmers in relation to the statement that each CSAP improves mitigation (reduce trees and vegetal cover destruction). Except for crop rotation (P3) and the use of funnel and animal droppings (P25), a high frequency of farmers disagreed with the statement that the practices improve mitigation.

## 4. Discussion

### 4.1. Climate Change Perception

In the study area, farmers have noticed changes in temperature, rainwater quantity and rainfall pattern. They have also heard about climate change through radio and NGOs. This means that climate change is already noticeable in the area and radio and NGOs play a role in raising farmers awareness on the phenomenon, supporting earlier findings from Africa (southern Africa, Kenya, Ethiopia) [21–23] and other parts of the world [24]. Results from studies by the World Bank [25] showed that significant numbers of farmers in Africa believe that temperatures have already increased and that precipitation has declined. Through a better understanding of climate change risk perceptions, there is a scope to design climate services that more readily fit the specific decision contexts of the African continent [26]. Climate change perception was not linked to any socio-demographic characteristics.

### 4.2. Awareness of Climate-Smart Agriculture Practices and Determinants of Adoption

The list generated by the stakeholders encompasses thirty-one (31) Climate-Smart Agriculture practices. These practices are used for maize, sorghum, rice, yams, groundnut, millet, hungry rice, etc. The current study found five (5) more practices in this zone: using ash against insects' attacks, double ploughing, biopesticides, cultivation of less demanding (with relatively reduced water and nutrients need) vegetables and sowing under plant cover. This may be due to the fact that the current study generated its list from a workshop gathering farmers, NGOs and governmental agricultural services, while the FAO [14] uses a list from experts. A study by CIAT, FAO and ICRISAT [27] also found that short duration crop varieties, zai planting pits, agroforestry, mulching, soil water conservation, erosion control, crop rotations, intercropping and staggered/relay cropping are common in Benin. In other parts of West Africa (Ghana), Anuga et al. [28] found that most farmers use CSAP such as personal experience to predict weather events, reliance on radio/television to access weather information, minimum tillage, organic manure and afforestation.

Almost 99% of farmers use at least one CSAP. This finding agrees with Kpadonou et al. and Ngetich et al. [29,30], who found that a high proportion of farmers use at least one CSAP. Seven (07) CSAP were used by more than 50% of those who know them. Out of the 31 CSAP, up to 24 practices have a low rate of use (less than 50% of those who know the practice use it). Among these are soil and water conservation practices. These findings support Kpadonou et al. [29], who found that in the Sahel region of Burkina Faso, the adoption of agricultural adaptation strategies was widespread, but specific adoption rates were very low for several practices. Even though the rate of use for a single practice is low, farmers adopt multiple CSAP. This is similar to the results by Kangogo et al. [31], who found that farmers adopt multiple CSA practices simultaneously and in combination as either complements or substitutes.

The adoption rates in this study are related to the percentage of those who are aware of the practice, which means that if the entire sample is considered, there would be lower rates of use. The first reason for not using CSAP is therefore ignorance of the practice. Awareness, and in the case of CSAP, appropriate information, is key to the decision to use a new technology. An important proportion of people does not know the practices. The choices of strategies are also determined by climate information, and farmers who are

aware of changes in climate are more willing to explore adaptation strategies [22]. A low rate of surveyed farmers adopted the CSAP associated with pest and disease attacks. This result confirms previous studies in Tanzania [32] reporting low adoption rates for CSAPs related to soil fertility, pest and disease attacks.

Some relatively well-known CSAP have a low rate of adoption. Surveyed farmers explained that they do not use direct parking even though they know the practice just because they do not have oxen. The spiritual invocation of rain requires the user to be initiated to a specific spirituality, and farmers do not use this practice either because they do not practice animism or they are not initiated to the rituals. Regarding the use of ash to control pests, farmers think this practice is less effective than the use of insecticide, resulting in a low rate of use for this practice.

The results of the binomial generalized mixed effect models revealed that ethnic group and education level affect the use of some practices. This is consistent with some author's findings (Anuga et al., Debela et al., and Ngetich et al. [23,28,30]) who reported that education was a determinant of CSAP adoption. The adoption of CSAP was not affected by location, age, household size and whether farmers belong to a group. This is contrary to Ngetich et al. and Lam et al. [30,33]. In the study area, ethnic groups are in specific areas, and the landscape is such that some ethnic groups are surrounded by mountains and crop on mountains. People living on mountains do not use low land because they do not have access to it. Some ethnic groups also find their soil to be hard to be left without ploughing and do not adopt no-tillage CSAP. Oxen are mostly owned by Fulani. People who park oxen on their farms often accept to collaborate with Fulani. In the study area, there are historical conflicts between Fulani and farmers and some ethnic groups do not collaborate with Fulani to have their animals parked in the farms.

Educated farmers do not believe that the spiritual invocation of rain is effective. They understand that using insecticide for edible crops is bad for human health and use ash for their gardens.

The three main constraints to agricultural production raised in this study are the lack of land, the lack of water and poor soil quality (cited by 78.7%, 71% and 90.2% of surveyed farmers, respectively). In addition, most interviewees were able to detect changes in the main climate parameters and have heard about climate change. In such conditions, it would be expected that a greater number of farmers adopt CSAP options such as water management and soil fertility. This was not the case, and this can be explained by the observed knowledge gap regarding CSAP determined by the level of education. Hence, there is a need for appropriate information concerning CSA options and their potential benefits with regard to crop productivity and the changing climate in a convenient language for the farmers with a low level of formal education to make the decision to adopt. The sound implementation of CSA options requires the definition of innovative policies and appropriate financial mechanisms to catalyze new initiatives that will ensure large-scale CSA adoption [2].

### 4.3. Evaluation of Climate-Smart Agriculture Practices

Two of the practices with high frequency of agreement with regard to CSA pillars were also found to be among the five best practices of 25 practices by the FAO [14] in Benin and according to expert evaluation. These are: improved crop varieties and rotation system. CSAP evaluation included in the FAO study [14] in Benin was performed by experts who can scientifically estimate the contribution of practices to each CSA pillar. In this study and similar to Manda et al. [34], CSAP were evaluated by farmers who have experience using the practices. Manda et al. [34] found in Tanzania that composting and improved drought-tolerant varieties contribute significantly to food security through their ability to increase productivity while ensuring adaptation to climate variability and change. This was similar to the current study, revealing that the use of improved varieties was classified by farmers among the best practices for all the three pillars.

Farmers agreed with the statements that practices improve productivity and adaptation except for "no-tillage". However, most farmers disagreed with most statements that CSAP contribute to mitigation. This may be because it is not easy to understand and evaluate mitigation. Whenever the link between a practice and deforestation is not clear, farmers fail to perceive mitigation. This is supported by the high frequency of farmers answering they do not know whether the practice reduces or not does not reduce greenhouse gas emissions. According to Manda et al. [34], productivity and adaptation are observable, estimable and deemed the most important by farmers.

## 5. Conclusions

The current study assessed smallholder farmers' perception of climate change and Climate-Smart Agriculture (CSA) and the determinants of adoption of CSA practices (CSAP) in Benin. The results revealed that in northern Benin (AEZ IV), farmers are aware of climate change. Up to thirty-one (31) CSAP were inventoried in the area, but most were not known by many farmers. The use of CSAP was determined by education level and ethnic group. Even though most farmers are aware of climate change and could enumerate constraints linked to soil quality and water and land availability in the area, only a few of them use CSAP. This may be the result of lack of knowledge that the CSAP exists or that the CSAP is effective. Most farmers agreed with the statement that CSAP increase crop productivity and income and contribute to adaptation for almost all CSAP but disagreed with the statement that CSAP contribute to climate change mitigation. The reason for such a result may be that mitigation is complex to explain or understand, while productivity and adaptation is relatively simple. Training on CSAP will help farmers to better understand essential CSAP and how CSAP contribute to each CSA pillar. This is critical for the adoption of CSAP and will contribute to the increase the area's resilience to climate change.

With many interventions of NGOs introducing a variety of technologies and practices in the study area, there was so far no published study that analyzes farmers' adoption of CSAP in the study area. In addition, contrary to previous studies that evaluated CSAP through experts, this work provides results of evaluation of CSAP by farmers, and this gives an understanding of how the final users of CSAP perceive the practices. Nevertheless, this study has one major limit. The study worked with a limited number of socio-demographic characteristics which did not allow to have a larger analysis of determinants. Based on this, we recommend that further research be conducted with more socio-demographic characteristics and an emphasis on the reasons for not adopting these practices.

**Author Contributions:** F.T.M. and G.T.T.-Y. conceived the research and implemented it. B.G., C.A., H.S. and M.T. substantially reviewed and edited the article. All authors have read and agreed to the published version of the manuscript.

**Funding:** This research was funded by Accelerating Impacts of CGIAR Climate Research for Africa (AICCRA) project. The AICCRA project is supported by a grant from the International Development Association (IDA) of the World Bank under grant number (AICCRA, P173398).

**Institutional Review Board Statement:** Not applicable.

**Informed Consent Statement:** Informed consent was obtained from all subjects involved in the study.

**Data Availability Statement:** Not applicable.

**Acknowledgments:** We acknowledge the funding received from the World Bank to the Accelerating Impacts of CGIAR Climate Research for Africa (AICCRA, P173398) project. The AICCRA project is supported by a grant from the International Development Association (IDA) of the World Bank. IDA helps the world's poorest countries by providing grants and low to zero-interest loans for projects and programs that boost economic growth, reduce poverty, and improve poor people's lives. IDA is one of the largest sources of assistance for the world's 76 poorest countries, 39 of which are in Africa. Annual IDA commitments have averaged about \$21 billion over circa 2017–2020, with approximately 61% going to Africa.

**Conflicts of Interest:** The authors declare no conflict of interest.

## Appendix A

**Table A1.** Description of CSA as provided by stakeholders.

| Number | Practice Name | Description |
|---|---|---|
| 1 | Spiritual invocation of rain | Causing rain to fall using spiritual practices |
| 2 | Climate risk forecast/early warning system | Traditional ways of knowing that a climate-related extreme event will come. Rural people use animals, nature observation to foretell events such as rain season, flood, drought, etc. |
| 3 | Crop rotation | Growing different crops successively on a piece of land |
| 4 | Sowing spread over time | Sow at different but close dates |
| 5 | Reduction in seed density | Increase the space between two sowing pockets so that the density is reduced |
| 6 | Use of improved varieties | Use of scientifically selected crop varieties |
| 7 | High number of seeds per pocket followed by thinning | Put more seeds in the pocket |
| 8 | Irrigation | Use of water cans or normal irrigation systems to provide water to the crop |
| 9 | Cropping on lowlands (marsh), flooded areas and riverbanks | Cropping on lowlands (marsh), flooded areas and riverbanks |
| 10 | Half moons | Cropping in a basin dug to form a half circle of 3 m diameter and where one mixes ground and manure |
| 11 | Ploughing perpendicular to the slope | Ploughing perpendicular to the slope |
| 12 | Fascine | Fence made of tree branches to protect gardens near the house |
| 13 | Use of Zai | Digging pits before the rain season to catch water when rain comes and concentrate compost |
| 14 | Staggered plowing | Plough in such a way that the next ridge's head blocks the furrow between the two ridges above |
| 15 | Stone rows | Use of stones to create small flat area on a sloppy area |
| 16 | Double plowing | Plough twice before sowing in order to destroy grass |
| 17 | No-tillage | No tillage = No tilling of the soil |
| 18 | Mulching | Use of decaying leaves or bark over the soil or around the crop |
| 19 | Direct seeding under cover | Sow directly the seed in a vegetation cover |
| 20 | Association of crops with Pigeon pea (*Cajanus cajan*) | Association of crops with Pigeon pea (*Cajanus cajan*) on the same plot |
| 21 | Association with *mucuna or Aeschynomene histrix, Stylosanthes guianensis* | Association with *mucuna or Aeschynomene histrix, Stylosanthes guianensis* on the same plot |
| 22 | Association Yam and Gliricidia | Association Yam and *Gliricidia* on the same plot |
| 23 | Cultivation of less water-intensive vegetables | Growing of vegetables that require less water than the others |
| 24 | Use of agricultural residues | Agricultural residues are left in furrows and decay to enrich the soils |
| 25 | Animal feces | Poultry, goat and sheep dropping as fertilizer |
| 26 | Use of compost | Use of compost as a fertilizer |
| 27 | Direct parking of oxen in the fields | Allow raisers to park their animals in the farm and gain from their feces as soil fertilizers |
| 28 | Use of chemical fertilizers | Use of chemical fertilizers |
| 29 | Use of micro-dose | Apply lower dose of chemical fertilizer at sowing in the pocket or at the base of plants two weeks after planting |
| 30 | Use of biopesticides | Use of neem tree oil, garlic, pepper and other botanical extracts and/or biological pesticides to control insect attacks on crops |
| 31 | Using Ash Against Attacks | Use ash to control insect attacks on crops |

## Appendix B

**Table A2.** Results of the generalized mixed effect model showing the relationships between farmers ethnic groups and the adoption of CSAP for significant factors.

| | Relationship between Ethnic Group and P7 | | | | Relationship between Ethnic Group and P9 | | | | Relationship between Ethnic Group and P17 | | | | Relationship between Ethnic Group and P27 | | | | Relationship between Ethnic Group and P31 | | | |
|---|---|---|---|---|---|---|---|---|---|---|---|---|---|---|---|---|---|---|---|---|
| | Estimate | Std. Error | z Value | Pr (>\|z\|) | Estimate | Std. Error | z Value | Pr (>\|z\|) | Estimate | Std. Error | z Value | Pr (>\|z\|) | Estimate | Std. Error | z Value | Pr (>\|z\|) | Estimate | Std. Error | z Value | Pr (>\|z\|) |
| Intercept | 1.9459 | 1.069 | 0.167 | 0.06872 | 1.95 | 1.07 | 1.82 | 0.0687 | $-2.88 \times 10^{-1}$ | $7.64 \times 10^{-1}$ | $-0.377$ | 0.70642 | $-2.52 \times 10^{-7}$ | $8.17 \times 10^{-1}$ | 0.00 | 1.0000 | $6.93 \times 10^{-1}$ | $8.66 \times 10^{-1}$ | $8.00 \times 10^{-1}$ | 0.4235 |
| Beberibe | 0.2513 | 1.5013 | 0.167 | 0.86706 | $1.90 \times 10$ | $8.70 \times 10^3$ | 0.002 | 0.9983 | 1.81 | $9.09 \times 10^{-1}$ | 1.995 | 0.04608 * | $-2.49$ | 1.10 | $-2.261$ | 0.0238 * | $-1.39$ | 1.22 | $-1.132$ | 0.25767 |
| Berba | $-1.0116$ | 1.1267 | $-0.898$ | 0.36927 | $-1.05$ | 1.12 | $-0.934$ | 0.3501 | 1.05 | $8.30 \times 10^{-1}$ | 1.266 | 0.20566 | $-5.23 \times 10^{-1}$ | $8.75 \times 10^{-1}$ | $-0.598$ | 0.55 | $1.67 \times 10^{-1}$ | $9.38 \times 10^{-1}$ | 0.178 | 0.85861 |
| Betamaribe | $-3.0204$ | 1.113 | $-2.714$ | 0.00665 ** | $-2.59$ | 1.11 | $-2.337$ | 0.0194 * | $-5.17 \times 10^{-1}$ | $8.18 \times 10^{-1}$ | $-0.632$ | 0.52741 | $-1.63$ | $7.02 \times 10^3$ | $-1.824$ | 0.0681 | $-1.25$ | $9.10 \times 10^{-1}$ | $-1.376$ | 0.16874 |
| Gourmatche | $-1.5404$ | 1.4058 | $-1.096$ | 0.27316 | $-1.54$ | 1.41 | $-1.096$ | 0.2732 | $6.93 \times 10^{-1}$ | 1.19 | 0.582 | 0.56032 | $-1.93 \times 10^1$ | $7.02 \times 10^3$ | $-0.003$ | 0.9978 | $-2.08$ | 1.41 | $-1.47$ | 0.14146 |
| Lopka | $-1.5953$ | 1.0849 | $-1.47$ | 0.14144 | $3.01 \times 10^{-1}$ | 1.11 | 0.272 | 0.7857 | $9.15 \times 10^{-1}$ | $7.89 \times 10^{-1}$ | 1.159 | 0.24645 | $-1.11$ | $8.40 \times 10^{-1}$ | $-1.319$ | 0.1871 | $-2.11$ | $8.98 \times 10^{-1}$ | $-2.353$ | 0.01863 * |
| Natimba | 0.3054 | 1.3021 | 0.234 | 0.81457 | $-1.03$ | 1.17 | $-0.878$ | 0.3801 | $7.73 \times 10^{-1}$ | $8.86 \times 10^{-1}$ | 0.872 | 0.38292 | $-9.16 \times 10^{-1}$ | $9.49 \times 10^{-1}$ | $-0.966$ | 0.3341 | $-2.94$ | 1.14 | $-2.58$ | 0.00988 ** |
| Waaba | 0.03774 | 1.1571 | $-0.183$ | 0.8551 | $2.03 \times 10^{-5}$ | 1.17 | 0.00 | 1.00 | 2.48 | $9.28 \times 10^{-1}$ | 2.68 | 0.00741 ** | $-2.51$ | 1.01 | $-2.479$ | 0.0132 * | $-2.64$ | $9.89 \times 10^{-1}$ | $-2.668$ | 0.00764 ** |

Significance codes: 0.001 '**' 0.01 '*'.

## Appendix C

**Table A3.** Results of the generalized mixed effect model showing the relationships between farmers' level of education and the adoption of CSAP for significant factors.

| | Relationship between Farmers' Education Level and P1 | | | | Relationship between Farmers' Education Level and P5 | | | | Relationship between Farmers' Education Level and P31 | | | |
|---|---|---|---|---|---|---|---|---|---|---|---|---|
| | Estimate | Std. Error | z Value | Pr (>\|z\|) | | Estimate | Std. Error | z Value | Pr (>\|z\|) | | Estimate | Std. Error | z Value | Pr (>\|z\|) |
| (Intercept) | $-0.6618$ | 0.2465 | $-2.684$ | 0.00727 ** | (Intercept) | 0.360815 | 0.254762 | 1.416 | 0.1567 | (Intercept) | $-0.5508$ | 0.1794 | $-3.071$ | 0.00213 ** |
| Primary school | $-0.7926$ | $-0.7926$ | $-2.343$ | 0.01912 * | Primary school | $-0.69462$ | 0.301715 | $-2.302$ | 0.0213 * | Primary school | $-0.6469$ | 0.337 | $-1.92$ | 0.05489 |
| Secondary school | $-0.6769$ | 0.3324 | $-2.037$ | 0.04170 * | Secondary school | 0.002403 | 0.297516 | 0.008 | 0.9936 | Secondary school | $-0.7614$ | 0.3505 | $-2.172$ | 0.02983 * |
| University level | $-19.9169$ | 6998.0538 | $-0.003$ | 0.99773 | University level | 0.957394 | 0.595625 | 1.607 | 0.108 | University level | $-0.7485$ | 0.6756 | $-1.108$ | 0.26792 |

Significance codes: 0.001 '**' 0.01 '*'.

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
