# Peer review of "Farmers’ Perception of Climate Change and Climate-Smart Agriculture in Northern Benin, West Africa"

_agronomy, doi:10.3390/agronomy12061348_

Round 1

Reviewer 1 Report

I found the article very interesting, since it tries to generate knowledge on an important topic, such as the adoption of agricultural practices that help mitigate the effects of climate change. In addition, it is very interesting to propose this study in an area where most farmers lack sufficient information and education to understand these issues, compared to farmers in other areas of the planet such as Europe or the US, where farmers have more resources and access to information. I congratulate the authors for the work.

However, I want to make a series of comments for the authors, with issues that I think could be improved in the article and some errors that I have found:

  1. The introduction seems scarce to me. I think the author should talk about some more current works related to the subject and similar, that have been carried out in Africa or in other parts in the world.
  2. Figure 2, the part on the right above is not clearly seen, where other areas of Africa are shown, the letters are crowded and cannot be distinguished. I think it would be better to place a map of all of Africa there and mark the study area, to better show the location of the study area on the continent.
  3. Perhaps it would be interesting to also know the most common causes or problems among the respondents for not adopting the best-known CSAP practices. That is, to know for sure the reasons why the best-known practices have relatively low rates of use. Asking questions to find out why farmers do not adopt the practice despite knowing about it.
  4. In line 261 it is said that the practice of spiritual invocation of rain is taken into account, although the percentage of knowledge by the respondents was less than 50% (48.9%), but in reality in the tables 1 and 2 show a percentage of knowledge of this practice of 59.5%. I think this could be a typo. If so, the authors should correct it.
  5. In my opinion, I think it is not necessary to show table 2, since it means repeating the content of table 1. Perhaps it would be enough to name these practices in a paragraph of text (which is already done in part between lines 252-257 and This table could be eliminated, which in my opinion does not add anything to the article since it repeats information and data that already appear in table 1).
  6. In my opinion, I think it is not necessary to use hyphens to cut words that don't fit at the end of one line and continue on to the next. I think this can be avoided by justifying the text. I recommend removing the hyphens and not cutting the words, because this contributes to a better and easier reading of the document.

Reviewer 2 Report

I reviewed the manuscript titled ‘Farmers’ perception of climate change and climate-smart agriculture in Northern Benin, West Africa’ with great interest. Its subject matter is most relevant, and it is well-structured. I do, however, have some comments and reservations that I believe need addressing to improve it.

Main questions:

  1. The main concern has to do with the apparent de-emphasis that the authors seem to do with the determinants of adoption in the middle of the manuscript. The conclusion (Line408-409) shows a dual objective enumerating the perceptions of farmers and then the determinants of CSA practices and yet the results and discussion sections are largely limited on the latter. The authors do not present the results from the generalized mixed-effects model. This should be presented even if the factors identified are not statistically significant. This is especially so as they seem to rely on the model on this to make substantial and far-reaching claims (Lines 250-257) that farmers’ ethnic and educational levels affect their use of CSA practices. Any regression table to show the magnitude of the contribution of the different factors? On the other hand, If the authors, after analyzing the data, concluded that they would remove the determinants of the adoption component from the manuscript, they should also do so in the abstract and the conclusions and explicitly state so.
  2. Section 3.2 could be better structured; too many 1-2 – sentence paragraphs.
  3. Table 2; Why did the authors not include staggered sowing (an important adaptation and mitigation measure) in the practices? I ask this question given the above-average usage rate of the practice (52%) according to Table 1.
  4. Authors’ claim on Line 398 that ‘… composting was not known enough in the area’ seems to be contradicted when juxtaposed against Table 1 which shows significant use of mulching (P18), crop residues (P24), animal droppings (P25) and perhaps most importantly, use of compost itself (P26)? Is the threshold 50% knowledge and/or usage rate? If so, what is the basis of this threshold? Similarly, why was staggered sowing excluded from the selected determinants Table 2? 
  5. And finally, how did the authors explain the phenomenon of climate change in the local dialects for the farmers to adequately understand? How did the authors differentiate this from climate variability? The distinction is quite critical in the face of Line 210 through 218.

Minor comments:

  1. Line 3: ‘Northern-Benin’ should not be having a hyphen in between. Ditto for other section of the manuscript such as Lines 18, 110,
  2. Seems to be an extra space before ‘Binomial’ on Line 21.
  3. Replace ‘factors’ with ‘characteristics’ on Line 22.
  4. Extra space on Line 39 before ‘Without’ should be removed
  5. Replace ‘is’ with ‘being employed’ on Line 42.
  6. Figure 1: What is the source of the figure? Was it self-produced by authors or they adopted and adapted it from literature? Also,  insert ‘emissions’ after greenhouse gas’ on Line 104.
  7. ‘Sahara’ on Line 123 needs the article ‘the’ to precede it.
  8. Paragraphs between Lines 124 and 132 need re-structuring.
  9. Add ‘farm’ to ‘land’ to make ‘farmland’ on Line 127.
  10. Claims on Lines 128-130 need some citation for support.
  11. Line 169: Were the authors meaning to write strives?
  12. Line 220: what does ‘media’ mean? Mass media (eg radio and television)? Also, what ‘meetings’ are the authors referring to? Farmer group meetings?
  13. Extra spacing on Line 354 between ‘of’ and ‘agricultural’ can be removed.
  14. Line 366 reads awkward, please rephrase.
  15. Any plausible reasons for these contrary findings on Line 372-374?
  16. Line 379: replace ‘great’ with ‘greater’.
  17. Line 384: Is the use of the phrase ‘illiterate farmers’ used here in the sense that they lacked formal education? Please clarify otherwise you risk coming across as insulting to this group of people. This is also the reason why presenting the results of the regression is critical.

Reviewer 3 Report

Dear Authors,

The article discusses how farmers in Benin perceive climate change and identifies actions needed to transform and reorganize agricultural systems to effectively support agricultural development and ensure food security in the face of climate change. It is relevant and interesting topic, nevertheless, the paper needs improvement.

Keywords do not reflect what is in the text, what does Adoption mean? It has many meanings. E.g. - it has not been indicated that the study is related to agriculture.

The text under picture 1 is in a smaller font

Line 106: The rate of adoption can be influenced by the Socio-demographic characteristics of smallholder farmers and the nature of the practice. when a practice is adopted and is acknowledged to be effective, other farmers follow the example. Full stop or comma?

Line 147: The authors used the method - stratified random sampling. How Authors divided the population into strata. What was the criterion?

Line 149-152: How many people participated in the workshops and how are they organized (where, are they all in one place, how many people were in the groups, etc.?). It is interesting for readers, e.g. from Europe, how such workshops are organized in Benin. Please explain.

Line 172-173: transfer this part to the discussion.

Line 189-192: as above. Do not place literature review in the methods section. Provide the necessary information in the introduction or in the discussion.

Line 211: Authors wrote „More than nighty eight percent….” I think that… ninety-eight… English for proofreading.

Conclusion section should not only sum up, but also provide conclusions coming from the research. In the article are insufficiently described.

Reviewer 4 Report

The article explores an interesting and current issue. The article does not stand on a very high level but also does not contain any major errors. I have a few comments on the content:

  1. A higher % pecentage of use than knowledge the practice, is this not a mistake? What is the reason for this?
  2. I have doubts whether the technique of Spiritual invocation of rain should be included in the article.
  3. Table 2 being a fragment of Table 1 not very necessary.
  4. The most serious concern. The aim of the article is: assessing the relationships between some socio-demographic factors and the use of CSAP. The research on this aspect was summarised in one sentence. In my opinion, therefore, part 4.2 of the article should be expanded and these relationships described in more detail.
